# Efficient On-Off Keying Underwater Acoustic Communication for Seafloor Observation Networks

**Yan Yao** [1,2,3] 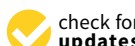, **Yanbo Wu** [1,3,4,*] , **Min Zhu** [1,3,4], **Dong Li** [1,2,3] **and and Jun Tao** [4,5,6]

1   Ocean Acoustic Technology Center, Institute of Acoustics, Chinese Academy of Sciences,
    Beijing 100190, China; yaoyan@mail.ioa.ac.cn (Y.Y.); zhumin@mail.ioa.ac.cn (M.Z.);
    lidong16@mails.ucas.ac.cn (D.L.)
2   University of Chinese Academy of Sciences, Beijing 100049, China
3   Beijing Engineering Technology Research Center of Ocean Acoustic Equipment, Beijing 100190, China
4   State Key Laboratory of Acoustics, Institute of Acoustics, Chinese Academy of Sciences,
    Beijing 100190, China; jtao@seu.edu.cn
5   Key Laboratory of Underwater Acoustic Signal Processing of Ministry of Education, Southeast University,
    Nanjing 210096, China
6   Acoustic Science and Technology Laboratory, Harbin Engineering University, Harbin 150001, China
*   Correspondence: wuyanbo@mail.ioa.ac.cn; Tel.: +86-130-1736-3111

**Abstract:** In the cableless seafloor observation networks (SONs), the links among network nodes rely on underwater acoustic communication (UAC). Due to the energy constraint and the high-reliability requirement of the cableless SONs, the noncoherent UAC has been a preferred choice, even though a noncoherent UAC scheme generally suffers from low spectral efficiency. In this paper, we propose a high-spectral-efficiency noncoherent UAC transmission scheme which is implemented as an orthogonal frequency-division multiplexing (OFDM) system adopting the on-off keying (OOK) modulation. To simultaneously achieve high performance at a low energy consumption, an irregular recursive convolutional code (IrCC) is employed and an accumulator (ACC) is introduced to achieve a modulation with memory at the transmitter side. The ACC enables a turbo iteration between the soft demapper called the ACC-OOK demapper and the soft decoder on the receiver side, and also reduces the decoding error floor. To account for the unknown signal-to-noise ratio (SNR), an iterative threshold estimation (ITE) algorithm is proposed to determine a proper decision threshold for the ACC-OOK demapper. The IrCC is designed to match the extrinsic information transfer (EXIT) curve of the ACC-OOK demapper, lowering the SNR threshold of the aforementioned turbo iteration. Simulations and experimental results verify the superiority of the proposed noncoherent UAC scheme over conventional ones.

**Keywords:** irregular recursive convolutional code; on-off keying; seafloor observation network; iterative threshold estimation

## 1. Introduction

The seafloor observation network (SON), as a real-time ocean observation platform, plays an essential role in the collection of ocean information, tsunami warning, positioning and navigation of vehicles, etc. So far, cabled SONs have been widely deployed, such as the Hawaii-2 Observatory (H2O) [1], the VENUS system deployed in Canada [2], the US NSF Regional Scale Nodes, and the NEMO-SN1 installed in Southern Italy [3]. The cables ensuring a high communication rate and sufficient power supply are expensive and inflexible though [4]. Cableless SONs connecting underwater sensors and instruments via acoustic links [2,5,6] therefore provide promising alternates for future seafloor observations.

### 1.1. Underwater Acoustic Communications for SONs

Underwater acoustic communication (UAC) schemes for several existing cableless SONs [2,5,6] are listed in Table 1, where both coherent and noncoherent ones can be found. Coherent schemes [2,5,7] have a high spectral efficiency, thus the potential to obtain a high data rate, e.g., 2.5–10 kbit/s was achieved with a bandwidth of 5–20 kHz [8]. However, they are vulnerable to harsh channel conditions and demand advanced receiver techniques and multichannel combining to guarantee transmission performance. By comparison, noncoherent schemes are much more robust despite the channel conditions [9] at the cost of degraded spectral efficiency. In the Deep-Ocean Assessment and Reporting of Tsunamis (DART) warning system [6], a noncoherent UAC using the multiple frequency-shift keying (MFSK) modulation was adopted. With only a single hydrophone on the receiver side, robust transmission was achieved at a low power consumption. It is noted by [8] that, in most SON applications, only short packets need to be transmitted periodically and the resulting data rate is no more than 1200 bit/s. Therefore, a low-power and high-reliability noncoherent UAC scheme is more desirable for a long-term cableless SON [10].

**Table 1.** Experiments or applications of the cableless SON.

| Application | Modulation | Distance | Bandwidth | Bit Rate | Number of Received Hydrophones |
|---|---|---|---|---|---|
| WHOI Experiment (2010) [2] | QPSK | 2 km | 22–27 kHz | 5300 bit/s | 4 |
| WHOI Experiment (2018) [5] | QPSK | 30 km | 3–4 kHz | 350 bit/s | 4 |
| The DART system (2013) [6] | MFSK | $\leqslant$7 km | 9–14 kHz | 600 bit/s | 1 |

### 1.2. Noncoherent UACs

The MFSK and on-off keying (OOK) are two typical noncoherent modulation schemes. In the present, the MFSK and its variants have been widely used in practice owing to their high robustness. The MFSK has a low spectral efficiency of $r = \log_2{(M)}/M$ bit/s/Hz with $M$ being the modulation level. The spectral efficiency reaches its maximum of $r = 0.5$ at $M = 2$ or 4 and is less than 0.5 when $M > 4$. The low spectral efficiency restricts the data rate [11] as the available bandwidth of a UAC channel is generally limited [12]. In [6], a data rate of 600 bit/s was achieved at a bandwidth of 5120 Hz. The corresponding spectral efficiency is only 0.12 bit/s/Hz. As another example, the frequency-hopping FSK(FH-FSK) was adopted in the WHOI Micromodem [13] and the Benthos telesonar modem [8]; both have a low data rate of about 80 bit/s. In contrast, the OOK modulation can achieve a higher spectral efficiency up to 1 bit/s/Hz, comparable to the coherent binary phase-shift keying (BPSK) modulation. For the OOK modulation, a carrier with fixed energy is transmitted only when a symbol is 'on.' Thus, it also has the advantage of lower-power consumption per bit, which enables a single-hydrophone receiver [14]. As a result, the OOK modulation is attractive for many applications in sensor networks [15]. However, there are still some challenges limiting the application of an OOK modulation scheme in SONs as follows:

- It is difficult determine an appropriate decision threshold for demapping [16], especially over the channel near the seafloor. Due to the severe fading and additive noise, the energy of each received symbol is no longer a deterministic value or zero;
- The conventional regular recursive systematic convolutional (RSC) codes have been widely employed as channel coding schemes in UACs. The signal-to-noise (SNR) threshold required for reliable decoding is generally high, which is unsatisfactory for SON applications.

### 1.3. Contributions

In this paper, we propose an efficient OOK UAC scheme for SONs. By addressing the aforementioned challenges, several contributions are made as follows:

- An accumulator (ACC) is introduced before a conventional memoryless OOK constellation mapping, leading to an OOK modulation with memory. It enables on the receiver a turbo iteration between the soft ACC-OOK demapper and the soft decoder and reduces the decoding error floor;
- An iterative threshold estimation (ITE) algorithm is proposed to seek an appropriate decision threshold for soft demapping and thus improve the iterative decoding performance;
- An irregular recursive convolutional code (IrCC) is designed by matching the extrinsic information transfer (EXIT) curve of the soft ACC-OOK demapper, so as to lower the SNR threshold of the turbo iteration;
- The proposed noncoherent UAC scheme is verified by both simulations and undersea experiments. It achieves a gain of 1.8 dB compared with a conventional scheme, and is only 3 dB to the channel capacity. In that sea trial at a communication distance of 2500 m, reliable transmission is achieved at a data rate of 1400 bit/s with a frequency band 6–10 kHz, i.e., the spectral efficiency is about 0.35 bit/s/Hz.

### 1.4. Relation to Existing Works

In our previous work [17], the IrCC [18] was introduced to UACs with some preliminary results presented. In this paper, a more extensive investigation is made. First, the complexity of the ITE algorithm is significantly reduced; second, the effects of iteration number and interleaver depth on the performance are thoroughly investigated; third, the placement of the communication nodes near the seafloor is analyzed; finally, the IrCC design is continuously improved by accounting for actual channel information.

The rest of this paper is structured as follows. In Section 2, we introduce the system model and the high spectral efficiency of the OOK modulation is demonstrated. In Section 3, the IrCC is designed and compared with the RSC, and the ITE algorithm for demapping is proposed. The Undersea application and experimental results are shown in Sections 4, and Section 5 concludes the paper.

## 2. System Model

### 2.1. System Description

The block diagram of the proposed noncoherent UAC scheme is shown in Figure 1. At the transmitter, the information bits are first encoded via a pre-designed IrCC encoder and then interleaved and fed to an ACC. The ACC is a unit-rate code encoder with the generating polynomial $g = 1/(1 + D)$. It introduces correlations among codewords without increasing their length, leading to an OOK modulation with memory. Attributed to the ACC, turbo iteration between the decoder and the demapper is thus enabled on the receiver side [19]. The IrCC is designed by matching the EXIT curve of the ACC-OOK soft demapper over the Rayleigh fading channel. The OOK symbols are partitioned into blocks, and each block is passed through an inverse fast Fourier transform (IFFT) module to obtain an orthogonal frequency-division multiplexing (OFDM) symbol. Guard intervals are inserted among OFDM symbols to avoid inter-block interference.

At the receiver, OFDM demodulation is performed via an FFT operation. After that, iterative detection is performed. The iterative detector consists of of a component ACC-OOK demapper and a component IrCC decoder. At each iteration, an LLR bias is evaluated based on the proposed ITE algorithm, as will be detailed in Section 3. Meanwhile, the channel fading information collected during the demodulation process is utilized to update the IrCC design, which is fed back to the transmitter for the next transmission. It is noted that: (1) the IrCC redesign is performed only when the channel fading characteristics undergo considerable change (e.g., every 3–4 h); (2) only subcode weighting coefficients of the redesigned IrCC need to be fed back and the involved payload is merely 200 bits. Hence, this feedback transmission can be implemented via a noncoherent UAC with MFSK modulation.

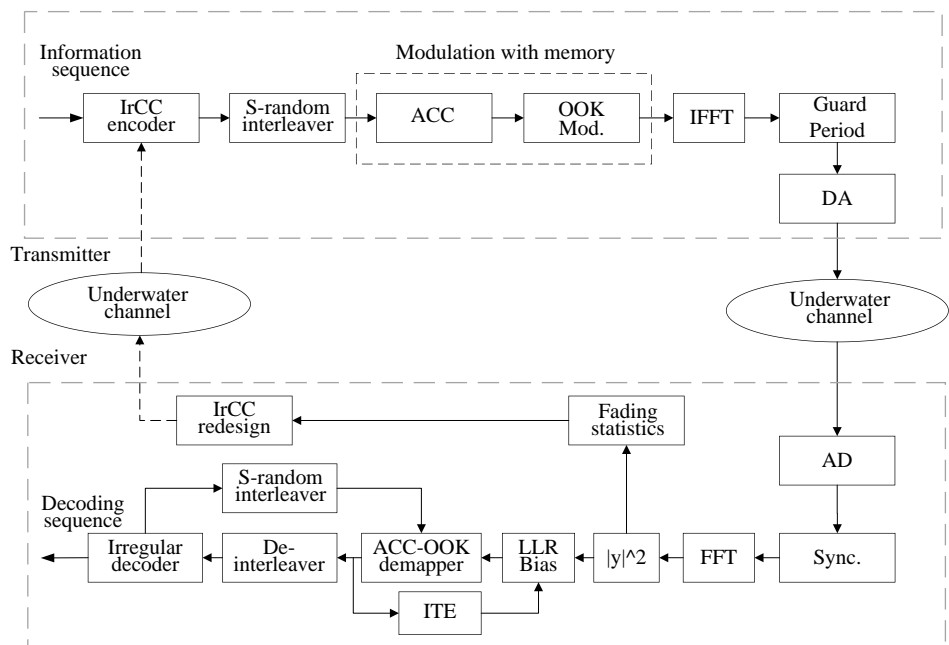

**Figure 1.** Block diagram of the proposed noncoherent UAC (underwater acoustic communication) scheme.

### 2.2. Channel Model and Capacity

The received signal at the $i$-th subchannel is expressed as

$$y_i = H_i u_i + n_i \tag{1}$$

where the channel fading coefficient $H_i$ and the additive white noise $n_i$ both are zero-mean circularly-symmetric complex Gaussian random variables with the variance of $2\sigma^2$ and $N_0$, respectively. The probability density function (PDF) of the magnitude of $H_i$, i.e., $|H_i|$, is thus Rayleigh distributed [20,21]. Without loss of generality, it is assumed $2\sigma^2 = 1$ in the following discussion. The transmitted symbol $u_i$ is taken from a binary OOK constellation. The conditional probability $P(Y_i|u_i)$ of the amplitude of the received signal, $Y_i = |y_i|$, is given as [22]

$$\begin{cases} p(Y_i|u_i = 0) = \frac{2Y_i}{N_0} \exp\left(-\frac{Y_i^2}{N_0}\right) \\ p(Y_i|u_i = 1) = \frac{2Y_i}{N_0+E_c} \exp\left(-\frac{Y_i^2}{N_0+E_c}\right) \end{cases} \tag{2}$$

where $E_c$ is the symbol energy. The channel capacity is computed as [21]

$$\begin{aligned} C &= E\left[\log_2\left(\frac{p(Y_i|u_i)}{p(Y_i)}\right)\right] \\ &= \frac{1}{2}\int_0^\infty \left[p(Y_i|0)\log_2\left(\frac{p(Y_i|0)}{p(Y_i)}\right)\right] dY_i + \frac{1}{2}\int_0^\infty \left[p(Y_i|1)\log_2\left(\frac{p(Y_i|1)}{p(Y_i)}\right)\right] dY_i \end{aligned} \tag{3}$$

where $P(u_i = 0) = P(u_i = 1) = 0.5$ is assumed and $p(Y_i) = \frac{1}{2}[p(Y_i|1) + p(Y_i|0)]$.

The capacity curves for four noncoherent modulations are compared in Figure 2, where $E_b/N_0$ is the SNR per bit. From the figure, two observations are made. First, the OOK can achieve a much higher spectral-efficiency than MFSK. Second, with the increase of spectral-efficiency, the $E_b/N_0$ threshold required by MFSK increases sharply. When the spectral-efficiency reaches the theoretically maximum value of 0.5 bit/s/Hz, the required $E_b/N_0$ threshold is infinite. By comparison, the OOK only requires $E_b/N_0 = 9.6$ dB for 0.5 bit/s/Hz.

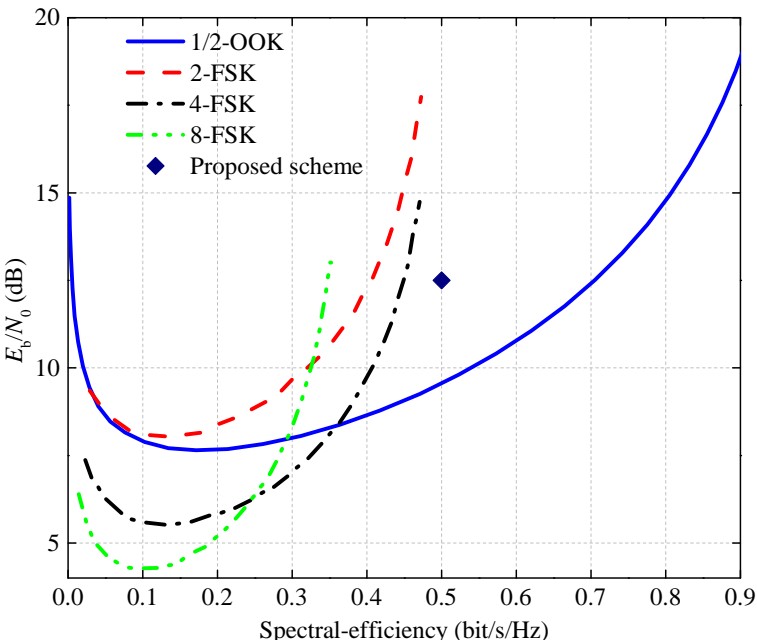

**Figure 2.** Capacity of four noncoherent modulations under flat Rayleigh fading channel without CSI on the receiver side.

## 3. Receiver Design Adapted to Channel

### 3.1. Iterative Threshold Estimation (ITE) Algorithm

Based on the Max-Log-APP algorithm, the a posteriori symbol LLR of the soft-input soft-output (SISO) ACC-OOK demapper can be expressed as

$$L(u_i) = L_c(u_i) + L_a(u_i) + L_e(u_i), 1 \leq i \leq K \tag{4}$$

where $L_c(u_i)$ is the channel observation LLR, $L_a(u_i)$ is the a priori LLR, $L_e(u_i)$ is the extrinsic LLR from the decoder. The $K$ denotes the length of an encoded sequence. The $L_c(u_i)$ is computed as

$$
\begin{aligned}
L_c(u_i) &= \ln \frac{p(Y_i|u_i = 1)}{p(Y_i|u_i = 0)} \\
&= \ln \left[ \frac{N_0}{N_0 + E_c} \exp \left( \frac{Y_i^2}{N_0} - \frac{Y_i^2}{N_0 + E_c} \right) \right] \\
&= \ln \frac{N_0}{N_0 + E_c} + \frac{E_c Y_i^2}{N_0(N_0 + E_c)} \\
&= A \left( Y_i^2 - B \right)
\end{aligned}
\tag{5}
$$

where $A = E_c / [N_0(E_c + N_0)]$ and $B = -\ln \frac{N_0}{N_0 + E_c} / A$ is an LLR bias. Due to the unknown SNR, the $L_c(u_i)$ cannot be directly computed based on Equation (5). To solve this dilemma, the ITE algorithm is proposed to determine $B$.

At the initial turbo-iteration, the channel observation LLR $L_c^0(u_i)$ is set as

$$L_c^0(u_i) = Y_i^2 - B^0 \tag{6}$$

where the $B^0$ is chosen as the median of $Y_i^2$, i.e., $B^0 = \text{median}\{Y_i^2, 1 \leq i \leq K\}$. For the $n$-th turbo-iteration of the decoding, the channel observation LLR $L_c^n(u_i)$ is computed as

$$L_c^n(u_i) = L_c^{n-1}(u_i) - B^n. \tag{7}$$

where $B^n$ is obtained by minimizing the hamming distance $D_H$ between the biased $\{L_c^n(u_i)\}_{\text{DE}}$ and $\{L_{\text{ACC}}^{n-1}(u_i)\}_{\text{DE}}$. The $L_{\text{ACC}}^{n-1}(u_i)$ is the output LLR of the ACC-OOK demapper from the previous iteration and $\{\cdot\}_{DE}$ represents the decision operation. Mathematically, $B^n$ can be obtained as

$$B^n = \arg\max_{B_j^n} \sum_{i=1}^{K} \text{sgn}\left[L_{\text{ACC}}^{n-1}(u_i)\left(L_c^{n-1}(u_i) - B_j^n\right)\right] \tag{8}$$

The search scope of $B^n$ depends on $\{L_c^{n-1}(u_i), 1 \leq i \leq K\}$ as

$$B_1^n, B_j^n, ..., B_K^n \in \left[\min\{L_c^{n-1}(u_i)\}, \max\{L_c^{n-1}(u_i)\}\right]$$

Intuitively, we can find an optimal $B^n$ from the whole search scope by sampling at equally intervals. For each sample $B_j^n$, the calculation of $\sum_{i=1}^{K} \text{sgn}\left[L_{\text{ACC}}^{n-1}(u_i)\left(L_c^{n-1}(u_i) - B_j^n\right)\right]$ is performed once. However, it is inefficient and has a total complexity of $\mathcal{O}(NMK)$ with $M$ being the number of samples ($M \leq K$). Instead, a low-complexity search algorithm is used. First, $\{L_c^{n-1}(u_i)\}$ is sorted into the ascending order yielding the sorted sequence $\{L_c^{n-1}(u_i)'\}$ which corresponds to a permutation sequence $P$. Based on $P$, the sequence $\{L_{\text{ACC}}^{n-1}(u_i)\}$ is also permutated into $\{L_{\text{ACC}}^{n-1}(u_i)'\}$. Then, an appropriate $B^n$ making the $D_H$ minimum can be directly chosen from the $\{L_c^{n-1}(u_i)'\}$. Details of the search algorithm are summarized in Algorithm 1. For each iteration, the main complexity of the ITE algorithm lies in the sort operation in step 4 and the summation in step 8, leading to the total complexity of $\mathcal{O}(NK\log K + NK)$.

---

**Algorithm 1** Search Algorithm of ITE

---

**Input:** $Y_i$; $\{L_{\text{ACC}}^{n-1}(u_i)\}$; $D^n, D_H^n$ are initialized as all 0 vectors of length $K + 1$; $D_l^n$ is the $l$-th element of the vector $D^n$; $\{\cdot\}_{DE}$ represents the decision operation.
**Output:** $B^n$
　1: **for** $n = 0 : N - 1$ **do**
　2:　　　$L_c^0(u_i) = Y_i^2 - \text{median}(Y_i^2)$　　// First LLR bias
　3:　　**if** $n \geq 1$ **then**
　4:　　　　　$[\{L_c^{n-1}(u_i)'\}, P^{n-1}] = \text{sort}(\{L_c^{n-1}(u_i)\})$　　// sort() is the sort function in MATLAB
　5:　　　　　$\{L_{\text{ACC}}^{n-1}(u_i)\}_{\text{DE}}' = \{L_{\text{ACC}}^{n-1}(u_i)\}_{\text{DE}}(P^{n-1})$　　// Permutation
　6:　　　　　$D^n(\{L_{\text{ACC}}^{n-1}(u_i)\}_{\text{DE}}' > 0) = 1$
　7:　　　　　$D^n(\{L_{\text{ACC}}^{n-1}(u_i)\}_{\text{DE}}' \leq 0) = -1$
　8:　　　　　$D_H^{i,n} = \sum_0^{l=i} D_l^n$　　// $D_H^{i,n}$ is the $i$-th element of the hamming distance vector $D_H^n$
　9:　　　　　$[D_H^{j,n}, j] = \min(D_H^n)$　　// The minimum hamming distance $D_H^{j,n}$ appears at the $j$-th
　　　　position
　10:　　　　　$B^n = \{L_c^{n-1}(u_i)'\}(j)$
　11:　　**end if**
　12: **end for**
　13: **return** $B^n$

---

### 3.2. IrCC Design under Rayleigh Fading Channel

In many SON applications, the RSC was adopted as the channel coding scheme due to its low complexity [6,8,13]. When the channel condition is good, it achieves decent decoding performance [23,24]. For a UAC channel with severe fading, however, it is no longer a good choice

due to the high required-SNR threshold for reliable decoding. In other words, it cannot adjust to the channel. In Figure 3, the iterative decoding process is characterized via the well-known EXIT chart [25], where $E_b/N_0 = 11.3$ dB and the RSC with the generating polynomial $(23,35)_8$ in octal indication is employed. The iterative tunnel between the EXIT curves of the ACC-OOK demapper and the soft decoder determines the performance and the convergence of the iterative decoding. Obviously, the iterative process is terminated in an early stage under the Rayleigh fading channel for the blocked iterative tunnel. To ensure an open tunnel, an increased $E_b/N_0$ is needed and this may not be available in energy-constrained SONs. Therefore, we choose to use an IrCC to lower the SNR threshold for obtaining an open tunnel.

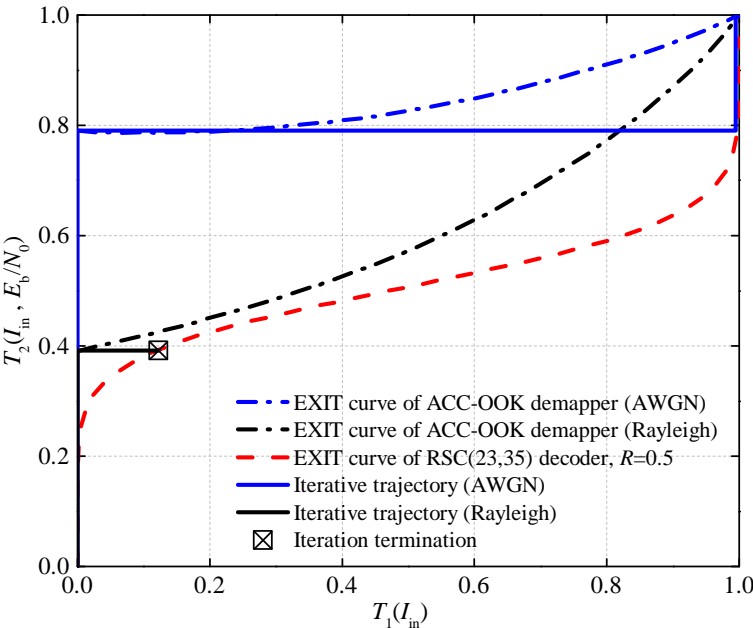

**Figure 3.** Premature terminated iteration of the system using RSC under the Rayleigh fading channel, $E_b/N_0 = 11.3$ dB.

The IrCC is designed as a weighted combination of 17 sub-codes with different rates $R_i = 0.1 + 0.05(i-1), i \in [1,17]$ [18], as shown in Figure 4. The information sequence of length $k$ is divided into segments of length $k_i$, which are respectively encoded by 17 sub-encoders, and then merged into the coding sequence. The 17 sub-codes are constructed by repeating and puncturing the mothercode defined by the generator $\frac{1}{g_0}[g_0, g_1, g_2, g_3]$, where

$$\begin{cases} g_0 = 1 + D + D^4 \\ g_1 = 1 + D^2 + D^3 + D^4 \\ g_2 = 1 + D + D^2 + D^4 \\ g_3 = 1 + D + D^3 + D^4 \end{cases} \tag{9}$$

To design an IrCC with target rate $R$ is equivalent to designing the weighting coefficients $\{\alpha_i\}$ for the sub-codes under the following constraint:

$$\begin{cases} \sum_{i=1}^{17} \alpha_i = 1, & \alpha_i \in (0,1) \\ \sum_{i=1}^{17} R_i \alpha_i = R, & R_i \in [0.1, 0.9] \\ \sum_{i=1}^{17} \alpha_i T_{1,i}(I_{in}) > T_2^{-1}(I_{in}, E_b/N_0), & I_{in} \in [0,1] \end{cases} \tag{10}$$

where $T_1\left(I_{\text{in}}\right)$ and $T_2\left(I_{\text{in}}, E_{\text{b}}/N_0\right)$ are the EXIT functions of the IrCC decoder and the ACC-OOK demapper, respectively. The distribution of $\alpha_i$ determines the shape of the IrCC decoding curve, thus affecting the iterative decoding performance. For a fixed code rate, the goal is to keep an open iterative tunnel until the $(1,1)$ endpoint. In other words, the IrCC decoding curve should match the ACC-OOK demapping curve, such that the $E_{\text{b}}/N_0$ threshold for reliable decoding will be effectively reduced.

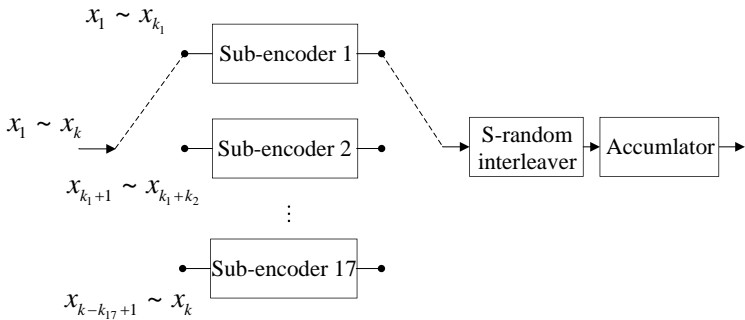

**Figure 4.** Demonstration of IrCC encoding.

The EXIT chart of the iterative process is shown in Figure 5, where the maximum mutual information optimized criterion [18] is adopted with a fixed code rate 0.5, and the optimized distribution of $\{\alpha_i\}$ is: $\{0.0012, 0.0018, 0.0028, 0.0059, 0.0178, 0.1105, 0.4690, 0.0329, 0.0338, 0.0057, 0.0254, 0.0527, 0.1035, 0.0653, 0.0294, 0.0270, 0.0153\}$. From the figure, the EXIT curve of the IrCC decoder matches that of the ACC-OOK demapper as expected, and thus the iterative tunnel is guaranteed, wherein the iterative trajectory deviates slightly from the iteration tunnel due to the limited interleaver depth [25].

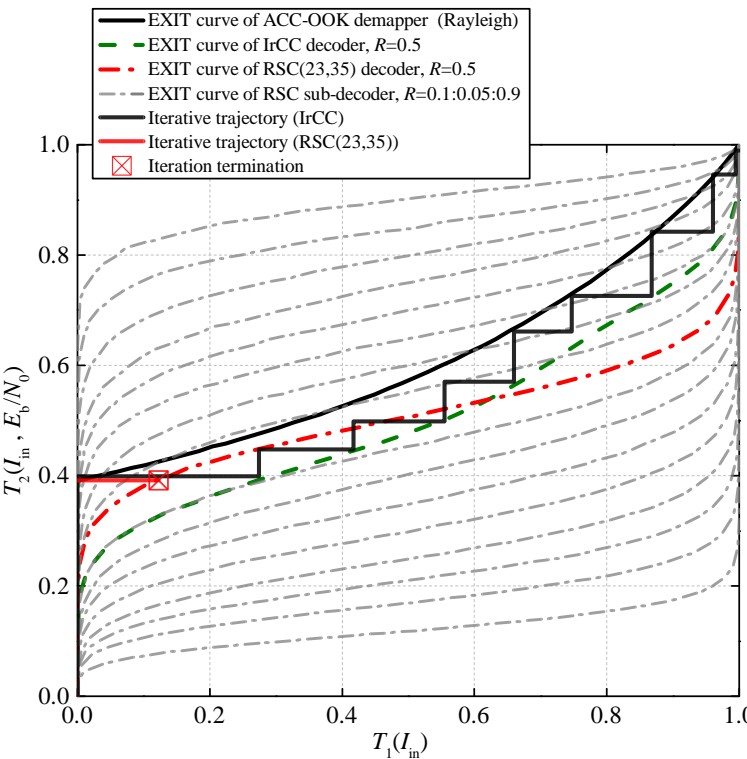

**Figure 5.** Iteration trajectory of the system using IrCC under the Rayleigh fading channel, $E_{\text{b}}/N_0 = 11.3$ dB.

In Figure 6, the bit error ratio (BER) performances of the IrCC and the RSC(23,35) are compared under the Rayleigh fading channel. The number of iterations is 10. Compared with the RSC, a gain of 1 dB is obtained by using IrCC at the BER level of $10^{-5}$. In Figure 7, the simulated performances of the proposed decoding scheme with ITE and a conventional decoding scheme without ITE under the flat Rayleigh fading channel are compared. Furthermore, we investigate the performance of the above three encoding and decoding schemes under different iterations. Three conclusions can be concluded. First, a performance gain of about 0.8 dB is achieved by adopting the proposed decoding scheme with ITE. Second, the decoding process of the RSC is almost unaffected by the number of iterations. In contrast, with the increase in the number of iterations, the decoding performance of IrCC is improved obviously, and the improvement is more significant with the aid of the ITE. Finally, at a maximum number of iterations of 20, reliable communication is achieved when $E_b/N_0 > 12.5$ dB, and the gap to the channel capacity is only 3 dB, as shown in Figure 2.

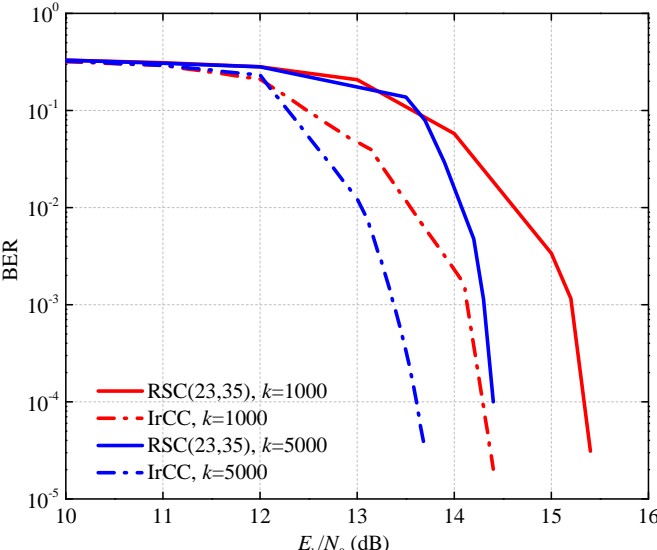

**Figure 6.** BER performance comparison of RSC and IrCC under the Rayleigh fading channel, $R = 0.5$.

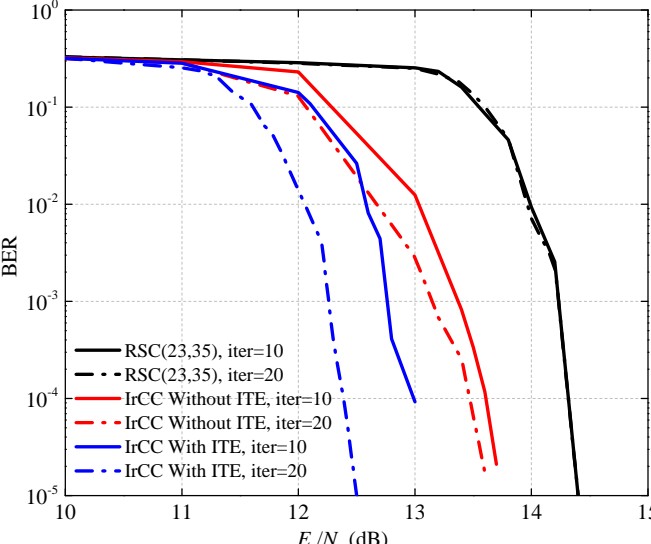

**Figure 7.** BER performance comparison of three encoding and decoding schemes under different iteration times, $R = 0.5$, $k = 5000$.

## 4. Undersea Application and Experimental Results

In this section, based on the SON application scenario, the seafloor communication experiment is described. As an important preliminary work, the placement of the communication nodes near the seafloor is analyzed to ensure good communication quality. Then, the feasibility and performance of the proposed scheme are verified by sea trial data.

### 4.1. Sound Rays Propagation and Placement of Communication Nodes

The speed of sound in the ocean is a function of temperature, salinity, and pressure, all of which vary significantly in the vertical [6]. These factors cause refraction, and the sound rays always bend in the direction where the sound speed decreases. Therefore, some areas where sound rays gather are called the convergence zone, while the others where no rays pass through are called the shadow zone. The communication nodes in the shadow zone would not be able to hear the transmitted signal. Due to the positive sound speed gradient, those sound rays always bend upwards when they are emitted from a source deployed near the seafloor. This phenomenon results in a large shadow area within a few kilometers of the source. The 20 sound rays with fixed launching angles ranging from −10 to 10 degrees are shown in Figure 8. The only different condition between Figure 8a,b is the height of the source from the seafloor. It can be concluded that, within a certain range of launching angles, the height of the source affects the range of the shadow zone and the number of the paths reflected by the seafloor. Moreover, the receiver hydrophone should be high enough from the seafloor to be in the convergence zone. In other words, the placement of the communication nodes directly affects the quality of communication. Moreover, the seafloor slope and communication distance should also be taken into account.

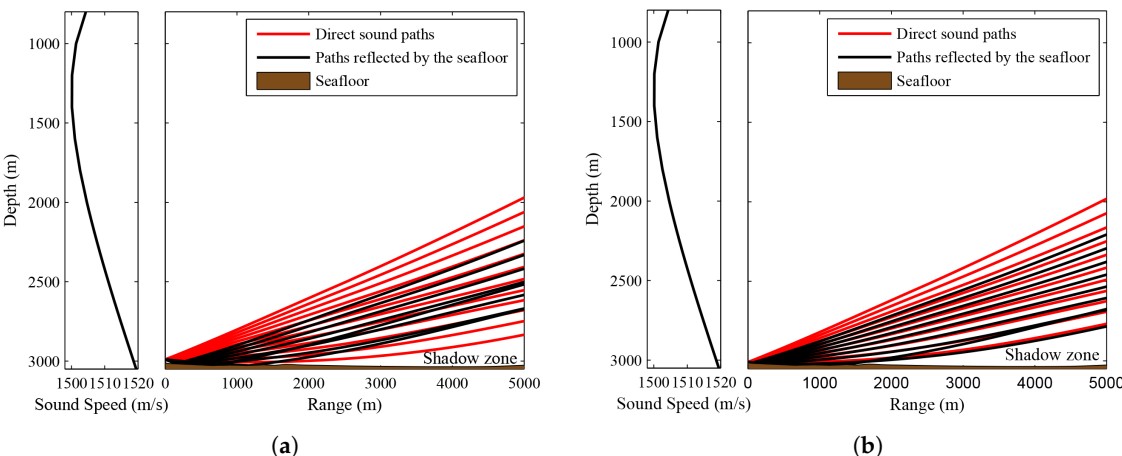

**Figure 8.** Sound ray paths from two sources with the heights of (**a**) 20 m and (**b**) 2 m away from the seafloor.

In this subsection, the constraint relation among the critical heights of the source and the receiving hydrophone, the seafloor slope angle, and the communication distance is analyzed. The propagation of the sound rays is modeled under ideal conditions. In order to simplify the calculation model, the typical deep-sea Munk model is adopted and the sound speed gradient near the seafloor is set to a constant *a*, which approximates the average sound speed gradient under the sound channel axis, and the seafloor topography is approximated as a flat floor with a certain slope angle *β*. The research object is the sound ray with the largest downward launching angle, which is the lower boundary of the optimal receiving region. This sound ray is assumed to be tangent to the seafloor in order to minimize the number of the paths reflected by the seafloor. According to the theory of ray acoustics, the sound ray in the case is an arc with equal curvature:

$$\rho = \frac{\mathrm{d}\theta}{\mathrm{d}s} = a \cos \alpha_0 \tag{11}$$

where $\theta$ is the direction angle of the target ray, $s$ is the length of the sound ray, $\alpha_0$ is the initial launching angle, i.e., the angle between the launching direction and the horizontal line, and $a$ is the relative speed gradient, which is defined as

$$a = \frac{c_i - c_{i-1}}{c_{i-1}(z_i - z_{i-1})} = 9.4 \times 10^{-6} \tag{12}$$

where $c_i$ is the sound speed at the depth $z_i$. Modeling, as shown in Figure 9, the target sound ray with the maximum launching angle $\alpha_0$ is tangent to the seafloor at point **A**, where **x** and **y** are the heights of the source and the receiving hydrophone from the seafloor, respectively. The point **B** is the projection of the receiving hydrophone on the seafloor.

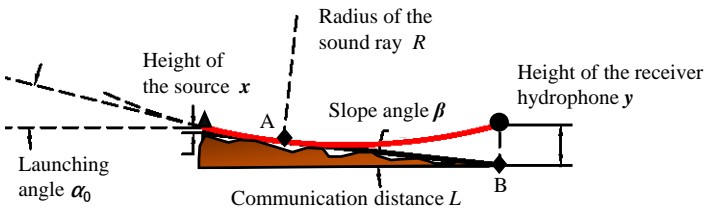

**Figure 9.** Schematic of the calculation model.

Taking the distance $z$ between the tangent point **A** and the projection point **B** as a parameter, the relationship between **x**, **y**, $L$ and $\beta$ is obtained by analytical calculation as follows:

$$\begin{cases} x(z,\beta) = \frac{2R}{\cos \beta} \sin^2 \frac{1}{2} \left[ \left( \arcsin \left( \frac{\left( \frac{L}{\cos \beta} - z \right) \cos \beta}{R} \right) - \sin \alpha_0 \right) + \alpha_0 \right] \\ y(z,\beta) = \frac{2R}{\cos \beta} \cos^2 \frac{1}{2} \left[ \left( \pi - \arcsin \left( \frac{z \cos \beta}{R} \right) - \sin \beta \right) - \beta \right] \\ z \in \left( max \left( 0, \frac{L - R \sin \alpha_0}{\cos \beta}, \frac{L}{\cos \beta} \right) \right) \end{cases} \tag{13}$$

*4.2. Experimental Results*

In June 2017, the UAC gateway and the observation node were deployed in the South China Sea, as shown in Figure 10. The depth of the seafloor was 1750 m. The heights of the gateway node and the observation node from the seafloor were 2 m and 140 m, respectively. Both nodes employed a single-hydrophone receiver and their horizontal distance was 2.5 km. The communication bandwidth was 6–10 kHz, which was divided into 120 subcarriers for transmission. For each block, the number of information bits is 60, and the duration of the cycle prefix was 10 ms. The observation node is equipped with two sets of batteries, which, together with power consumption under different working modes, are listed in Table 2. When the update cycle is half an hour and the gateway works continuously, the battery power can last for one year.

The measured channel is normalized and further statistically counted in Figure 11, where the probability density function (PDF) of the measured channel energy (i.e., $|H_i|^2$) is shown with that of the Rayleigh fading channel as a reference. From the figure, the fading degree of the actual channel is lower compared with the Rayleigh fading channel. Thus, during the communication interval, the IrCC is redesigned according to the statistical results to match the real channel. The BER curve of the proposed scheme is drawn in Figure 12, where one observes that reliable communication is achieved when $E_b/N_0 > 9.2$ dB. The proposed ITE algorithm achieves a performance gain of 0.8 dB. The IrCC redesign based on the statistical results brings 1 dB performance gain to the system. The value of the strategy of designing codewords to match channel conditions has been proved. Based on the non-coherent UAC

system, the achieved rate is 1400 bit/s, i.e., the spectral-efficiency is 0.35 bit/s/Hz, which can satisfy the requirement of the real-time transmission between the sensors and the shore-based station.

**Table 2.** Observation node power consumption.

| Battery Types | Battery Capacity (kWh) | Working Mode | Power Consumption (W) |
|---|---|---|---|
| 3.3 V battery | 0.2 | Power control and acoustic wake-up detection | 0.01 |
| 48 V battery | 2.3 | Transmitting mode | 80 |
| | | Receive processing mode | 2 |

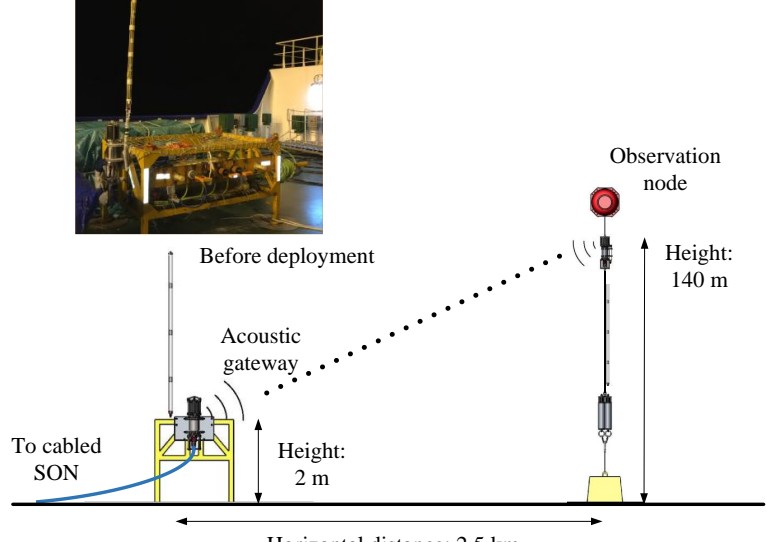

**Figure 10.** Demonstration of communication nodes in the SON experiment.

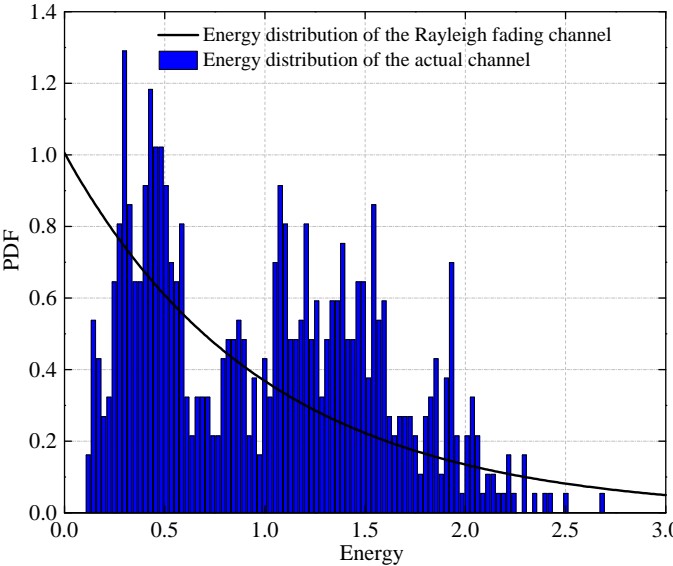

**Figure 11.** Normalized energy distribution of the actual channel.

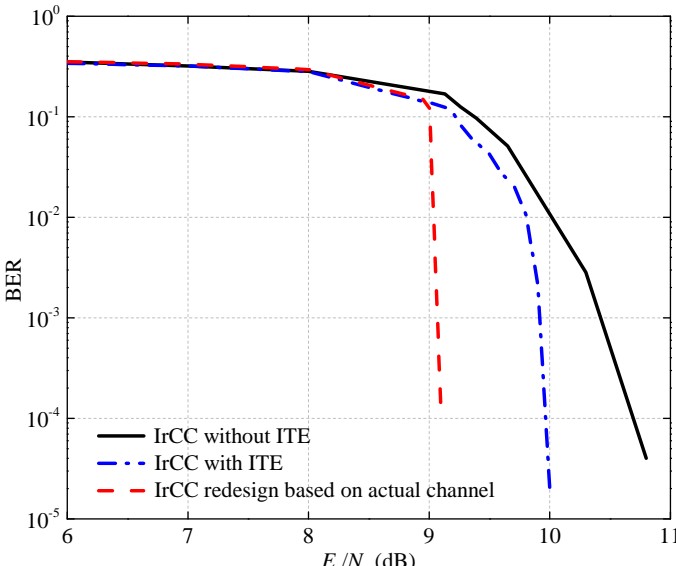

**Figure 12.** Performance comparison of three encoding and decoding schemes in the actual channel, $k = 1000$.

## 5. Conclusions

For the applications of the SON, a novel non-coherent UAC multi-carrier transmission scheme adopting the OOK modulation was proposed in this paper. Firstly, we verified the high spectral efficiency of the OOK modulation by calculating the channel capacity under the flat Rayleigh fading channel. The lower bound of $E_b/N_0$ is 9.6 dB when the capacity is 0.5 bit/s/Hz. To approach the channel capacity, an ACC was coupled with the OOK modulation to facilitate the turbo-iteration, and an IrCC was designed to match the EXIT curve of the ACC-OOK demapper and thus lowered the iterative SNR-threshold. A low complexity ITE algorithm was also proposed to improve the iterative decoding performance. The simulation results showed that, under the flat Rayleigh fading channel, the proposed scheme can achieve reliable communication when the $E_b/N_0 > 12.5$ dB and the gap to the channel capacity was only 3 dB. Compared with the conventional scheme adopting RSC without ITE, a performance gain of 1.8 dB was obtained. Furthermore, as an important preliminary work of sea trial, the placement of the communication nodes near the seafloor was analyzed to ensure good communication quality. Finally, the proposed scheme was verified by the SON experimental data. Reliable communication has been achieved when $E_b/N_0 > 9.2$ dB, and the achieved spectral efficiency was 0.35 bit/s/Hz with frequency band 6–10 kHz at the communication distance of 2500 m.

**Author Contributions:** Conceptualization, Y.Y., Y.W., and D.L.; methodology, Y.W. and M.Z.; software, Y.Y. and Y.W.; investigation, Y.Y. and Y.W.; resources, M.Z.; writing—original draft preparation, Y.Y., Y.W., D.L., and J.T.; writing—review and editing, Y.Y., D.L., and J.T.; visualization, Y.Y., Y.W., and D.L.; supervision, M.Z. All authors have read and agreed to the published version of the manuscript.

**Funding:** This work was supported in part by the National Natural Science Foundation of China under Grants 61971472, 61871114 and 61471351, in part by the Strategic Priority Research Program of the Chinese Academy of Sciences under Grant XDA22030101, in part by the National Key R&D Program of China under Grant 2016YFC0300300, in part by the Fund of Acoustic Science and Technology Laboratory, Harbin Engineering University, in part by the Open Funds of State Key Laboratory of Acoustics, Chinese Academy of Sciences, under Grant SKLA201805, in part by the China Scholarship Council, under Grant 201904910081, and in part by the Innovation Special Zone of National Defense Science and Technology of China.

**Conflicts of Interest:** The authors declare no conflict of interest.

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
