# Peer review of "Efficient On-Off Keying Underwater Acoustic Communication for Seafloor Observation Networks"

_applsci, doi:10.3390/app10061986_

Round 1
Reviewer 1 Report
This paper presents a very interesting application, which is highly suitable for the MDPI Applied Sciences Journal.
The application is novel and genuinely interesting. The authors have introduced an accumulator into an OOK modulation scheme to reduce errors within UAC-based environments. The iterative thresholding algorithm is used for improving decoding results. IrCC is used for channel coding derived from SNR informed threshold analysis with the Accumulator-OOK modulation and IrCC scheme. The results obtained demonstrate the advantages of the method implemented.
If possible, the authors should provide a comparison with other current published studies.
The English grammar requires significant edits to improve clarity and enable the paper to be of a publishable standard. Some sections have significant issues, such as introduction, while others, such as section 2 are much improved. As it currently stands the premises being presented are unclear at points, taking away the impact of the work and leading to confusing points throughout.
The article can be verbose and unspecific in presentation, leading to unclear discussion/statements.
Figure 1 raises questions into the IrCC redesign with channel fading estimation details communicated from the Receiver to the Transmitter. Is this through a fixed physical communication or wirelessly communicated back with the proposed scheme. This is a key aspect to the proposed scheme and should be explicitly detailed.
The Big-O complexity values are presented with no explanation for the obtained complexity from the proposed equations.
Undersea Application and Experimental Results are detailed in a excellent level of detail. The use of figures strongly aids in contextualising the contributions.
Author Response
We appreciate your careful review and valuable comments, which have significantly helped us to improve the quality of the manuscript. Next address your comments.
- This paper presents a very interesting application, which is highly suitable for the MDPI Applied Sciences Journal. The application is novel and genuinely interesting. The authors have introduced an accumulator into an OOK modulation scheme to reduce errors within UAC-based environments. The iterative thresholding algorithm is used for improving decoding results. IrCC is used for channel coding derived from SNR informed threshold analysis with the Accumulator-OOK modulation and IrCC scheme. The results obtained demonstrate the advantages of the method implemented.
- Answer: Thanks for your positive comment.
- If possible, the authors should provide a comparison with other current published studies.
- Answer: Thanks for the suggestion. We respectfully note that some published studies are compared in Table 1 in terms of technical indicators, and Section 1.4 has been added in the last revision to illustrate further the improvements we have made to the existing works. The idea of applying IrCC combined with OOK to noncoherent UAC proposed in this paper is groundbreaking to some extent, so the published works that can be compared are limited. We sincerely hope you will be considerate of this.
- The English grammar requires significant edits to improve clarity and enable the paper to be of a publishable standard. Some sections have significant issues, such as introduction, while others, such as section 2 are much improved. As it currently stands the premises being presented are unclear at points, taking away the impact of the work and leading to confusing points throughout. The article can be verbose and unspecific in presentation, leading to unclear discussion/statements.
- Answer: Thanks for the comments.
- Action: Some sections have been significantly edited for clarity. And we have reviewed the full text for many times and corrected the grammar errors we found.
Line 4-16, 24-94: Abstract and Introduction have been rewritten for clarity and brevity.
Line 107-132: System Description has been rewritten to make the presentation more accurate.
Line 138, 142, 147-149, 150, 152, 154, etc.: Some grammar and diction errors have been corrected, and some redundant descriptions have been removed.
- Figure 1 raises questions into the IrCC redesign with channel fading estimation details communicated from the Receiver to the Transmitter. Is this through a fixed physical communication or wirelessly communicated back with the proposed scheme. This is a key aspect to the proposed scheme and should be explicitly detailed.
- Answer: Thanks for the comment. The result of the irregular code redesign is fed back by underwater acoustic communication.
- Action: We have added an explanation of this process in the line 133-136 in Section 2.1. And Figure 1 is also improved.
- The Big-O complexity values are presented with no explanation for the obtained complexity from the proposed equations.
- Answer: Thanks for the comment. In the original manuscript, only the complexity of the search algorithm was compared, which may have caused some confusion.
- Action: The Big-O complexity value of the ITE algorithm is presented and explained in the line 173-174, 181-183 in Section 3.1
- Undersea Application and Experimental Results are detailed in an excellent level of detail. The use of figures strongly aids in contextualising the contributions.
- Answer: Thanks for your positive comment.

Reviewer 2 Report
The paper sounds quite interesting. The introduced technique is well described, and, in addition, the performance comparison among the considered techniques is explicative and well performed.
Author Response
Thanks for the comment.
- The paper sounds quite interesting. The introduced technique is well described, and, in addition, the performance comparison among the considered techniques is explicative and well performed.
- Answer: Thanks for your positive comment.

This manuscript is a resubmission of an earlier submission. The following is a list of the peer review reports and author responses from that submission.
Round 1
Reviewer 1 Report
In the paper the basic model of the modulated system is mistaken, as the conditional probability density function of the signal of (1) (here is the equation)
where H sub i is the fading coefficient and the n sub i is the zero-mean additive white noise both are zero-mean
complex Gaussian random variables with variance of 2 sigma square and N sub zero respectively is problematic.
Let us consider the conditional probability density function in the case when the condition is zero.
According to the equation (2) of the article: (here is the equation)
and is a complex random variable with for example kszi and eta real and imaginary components. Therefore (the equation is given here)
The conditional joint probability density function of kszi and eta is given as follows: (here is the equation)
, but it is not a conditional probability density function, as the double integral of it is not equal 1.

Reviewer 2 Report
In this paper, the authors propose a high spectral efficiency noncoherent UAC
3 multi-carrier transmission scheme adopting the on-off keying (OOK) modulation.
This paper is very well-written and organized. The subject is interesting to the research community and therefore, has archival relevance. My comments on the work are
1 - Please, provide a related work section where you contrast your work with the state of the art.
2 - Please, carefully review the text in order to correct grammar and typos.